# Mass Spectrometry and Pharmacological Approaches to Measuring Cooption and Reciprocal Activation of Receptor Tyrosine Kinases

**DOI:** 10.3390/proteomes11020020

**Published:** 2023-06-02

**Authors:** Jason Linzer, Zachary Phelps, Shivasuryan Vummidi, Bo Young Elizabeth Lee, Nicolas Coant, John D. Haley

**Affiliations:** Department of Pathology and Cancer Center, Stony Brook University, Stony Brook, NY 11794, USA

**Keywords:** receptor tyrosine kinase, signaling crosstalk, cancer therapy

## Abstract

Receptor tyrosine kinases (RTKs) can show extensive crosstalk, directly and indirectly. Elucidating RTK crosstalk remains an important goal in the clinical combination of anti-cancer therapies. Here, we present mass spectrometry and pharmacological approaches showing the hepatocyte growth factor receptor (MET)-promoting tyrosine phosphorylation of the epidermal growth factor receptor (EGFR) and other membrane receptors in MET-amplified H1993 NSCLC cells. Conversely, in H292 wt-EGFR NSCLC cells, EGFR promotes the tyrosine phosphorylation of MET. Reciprocal regulation of the EGFR and insulin receptor (IR) was observed in the GEO CRC cells, where inhibition of the EGFR drives tyrosine phosphorylation of the insulin receptor. Similarly, in platelet-derived growth factor receptor (PDGFR)-amplified H1703 NSCLC cells, inhibition of the EGFR promotes the tyrosine phosphorylation of the PDGFR. These RTK interactions are used to illustrate basic principles applicable to other RTK signaling networks. More specifically, we focus on two types of RTK interaction: (1) co-option of one RTK by another and (2) reciprocal activation of one receptor following the inhibition of a distinct receptor.

## 1. Introduction

Receptor tyrosine kinases (RTKs) are key mediators of tumor cell survival, proliferation, and migratory pathways [1,2], and inhibitors of RTKs have demonstrated anti-tumor efficacy in both the preclinical and clinical settings [3,4]. The quantitative measurement of cancer cell signaling under dynamic conditions of pharmacological or siRNA-mediated inhibition of specific signaling nodes provides insight into the requirements for effective cellular blockade of survival and invasive networks associated with cancer progression. Equally important, such studies can reveal mechanisms by which cancer cells can bypass and become resistant to RTK inhibitors, opening the potential for new drug target discovery and validation [5]. The use of quantitative shotgun LC-MS/MS methods incorporating stable isotope-labeling provides a rapid means to identify proteins, phosphoproteins, and specific phosphorylation sites perturbed in cancer cell states and conditions. Importantly, due to non-specific binding events common to affinity chromatography approaches, global mass spectrometry experiments involving very-low-abundance proteins or phospho-peptides can be ‘hypothesis-generating’, sometimes requiring additional validation steps, for example, via scheduled MRM, Western immunoblot, or ELISA approaches. However, these untargeted mass spectrometry methods have allowed interrogation of RTK crosstalk in tumor cell lines and xenografts and have generated relevant hypotheses related to clinically actionable drug combinations.

RTKs transduce extracellular growth and survival signals through multiple signaling nodes within the cell, three of which are the RAS-RAF-MEK-Erk/MAPK pathway, the JAK-STAT pathway, and the PI3K-AKT-mTOR pathway [1]. The MAPK, STAT, and AKT pathways promote increased cellular growth and survival, and parallel pathways activating these signaling nodes can mediate resistance to targeted therapy, chemotherapy, and radiation. Given their central role in tumorigenesis, RTKs have been targets of intense interrogation for the design of anti-tumor agents [3]. Although preclinical and clinical studies have demonstrated success for both antibody and small-molecule RTK inhibitors as single agents, acquired resistance is common, and extensive efforts have focused on identifying the molecular mechanisms that contribute to acquired resistance, with the aim of optimizing efficacy through patient selection and rational drug combination strategies.

Here, we examine the use of mass spectrometry in the co-option and reciprocal regulation of RTK signaling networks. We use the cell models of MET, IGF1R, EGFR, and PDGFR (Figure 1) as well as specific pharmacological inhibitors as examples. These RTKs can be activated by autocrine or paracrine growth factor binding, as illustrated in Figure 1, by specific mutations (e.g., EGFR exon 19 deletion or L858) and by gene amplification and overexpression, which increase receptor density on the cell surface, promoting ligand-independent kinase activation [4].

## 2. Methods

### 2.1. Cell Culture, Inhibitors, Antibodies, and Immunoblot

Non-small-cell lung cell lines H292, H1993, H1650, and H1703 were obtained from the American Type Culture Collection (ATCC, Manassas, VA, USA) and cultured in the prescribed media. Colorectal GEO cells were the gift of Dr. Michael Brattain. Cell lines were cryo-preserved and passaged less than six months. Inhibitors of EGFR (Erlotinib), MET (Crizotinib), and IGF1R/IR (Linsitinib) were obtained from MedChem Express (Monmouth Junction, NJ, USA). Antibodies directed to phospho-MET (Y1234, Y1235; #3077S), phospho-EGFR (Y1173S; #4407), beta-actin (#8457S), and PDGFRα (#3164S) were from Cell Signaling Technologies (Danvers, MA, USA). Antibodies to pPDGFR (Tyr 754, #12911), pPDGFR (Tyr720, #12910), and GAPDH (#25778) were from Santa Cruz Biotechnology (Dallas, TX, USA). Protein immunodetection was performed by electrophoretic transfer of SDS-PAGE-separated proteins to PVDF, incubation with antibody, and chemiluminescent second-step detection.

### 2.2. Phosphoprotein and Phosphopeptide Isolation

Anti-phosphotyrosine affinity fractions were isolated from Triton X-100 cell lysates as previously described [6,7,8]. Essentially cells were lyzed in 50 mM HEPES pH 7.5 containing 150 mM NaCl, 1.5 mM MgCl_2_, 1 mM EGTA, 10% glycerol, 1% Triton X-100, protease inhibitor cocktail (Sigma-Aldrich, St. Louis, MO, USA), and 1 mM sodium orthovanadate. Anti-phosphotyrosine antibodies, PY20 and PY100, were bound to Protein G-resin for 30 min at room temperature, followed by crosslinking with 5 mM disuccinimidyl suberate (DSS) for 1 h at room temperature. Non-covalently bound IgG was removed with 0.2 M sodium citrate pH 2.8. Antibody bead conjugates were then incubated with cell lysates for ~5 h at 4 °C with rotation. Antibody–antigen complexes were washed with 10 mM HEPES pH 7.5 and 150 mM NaCl at 4 °C, and bound proteins were then eluted with 0.1% TFA and 5% methanol in water and lyophilized. Anti-phosphotyrosine-isolated protein species were denatured in 0.5 M triethylammonium bicarbonate and 0.1% sodium dodecyl sulfate, reduced with 5 mM Tris-(2-carboxyethyl)phosphine (TCEP), alkylated with 10 mM methyl methanethiosul–-fonate (MMTS), and cleaved with trypsin (10 ug, 37 °C, 16 h). Peptide N-terminal α-amino and lysine ε-amino groups were labeled with isobaric iTRAQ tags [7]. Peptides were further purified by cation exchange (SCX) chromatography (4.6 × 5 mm, polysulfoethyl A), followed by C18 reverse-phase desalting.

Serine and threonine phosphorylated peptides were enriched by binding to titanium dioxide. Essentially, phosphopeptides were enriched on TiO_2_ beads (GL Bioscience, 10 u, Torrance, CA, USA) in 50% acetonitrile, 0.1% formic acid, and 1 M lactic acid for 90 min, RT with mixing. Beads were washed three times with 50% acetonitrile and 0.1% formic acid and eluted in 20 mM dipotassium phosphate pH 10.5 and 50% acetonitrile. Eluates were immediately neutralized by mixing with an equal volume of 5% formic acid. Phosphopeptides were desalted by C18 step chromatography, followed LC-tandemMS.

### 2.3. LC-MS/MS and Data Analysis Conditions

Anti-pY-enriched samples were analyzed by LC-MS/MS, followed by protein database searching. HPLC C18 columns were self-packed with 3 u Magic Aqua C18 resin (75 u ID × ~15 cm). Peptides were separated by 2–3 h gradient reverse-phase HPLC, starting with 0.1% formic acid, with increasing acetonitrile (0.39%/min) over 90 min. Electrospray ionization used a spray voltage of ~2.3 kV. Information-dependent MS and MS-MS acquisitions were carried out on an orthogonal quadrapole-TOF instrument (5600+ and QSTAR-Elite instruments; Sciex, Framingham, MA, USA) using a 0.2 s survey scan (m/z 400–1600), followed typically by 10 consecutive second-product ion scans (m/z 60–1200). Parent ions with charge states of 2+, 3+, and 4+ were selected. MS data were collected using Analyst. Mean mass accuracy for MS peptide scans was ~0.02 Da. Proteins were identified from survey and product ion spectra using the Paragon algorithm of ProteinPilot (Sciex, Framingham, MA, USA), searching human SwissProt and UniProt protein databases. Protein identification complied with the guidelines of Bradshaw [9], where 2 or more unique isoform-specific peptides were required for inclusion. Phosphopeptides were measured by determination of the median normalized iTRAQ quant tag peak areas.

Phosphopeptides isolated by TiO_2_ affinity were analyzed using an orbital trap instrument (Thermo Q-Exactive HF, Thermo Fisher, Waltham, MA, USA), followed by protein database searching. HPLC C18 columns (100 μm ID × ∼20 cm) were self-packed with 3 μm Reprosil C18 resin. Peptides were separated on the resolving column with a flow rate of 300 nL/min gradient elution over 2 h (0.1% formic acid/water—ACN). Electrospray ionization used a spray voltage of 2.3 kV. Data-dependent MS and MS-MS acquisitions were carried out using a survey scan (m/z 375−1400) with maximum fill of 50 ms, followed typically by 20 consecutive product ion scans (m/z 100−1600). Parent ions with charge states of 2+, 3+, 4+, and 5+ were selected with a 15 s exclusion period. MS data were collected using Xcaliber (Thermo Fisher, Waltham, MA, USA). Raw data were analyzed using Proteome Discoverer v2.2 software (Thermo Fisher, Waltham, MA, USA) using label-free quantitation. Resolution of MS and MS/MS data searches were set to 10 ppm and 0.05 Da, respectively. Two skipped trypsin cleavages and modifications of M-oxidation, KR-deamidation, ST-dehydration, and STY-phosphorylation were allowed. Peptide identifications were binned at <1% and <5% FDR cutoffs. A human UniProt dataset (42,252 entries) was used for data alignment. Fold change ratios were obtained by matched peptide-based label-free quantitation, and *p*-values were calculated by Benjamini–Hochberg correction for FDR.

## 3. Results and Discussion

### 3.1. Approaches for Defining and Measuring RTK Signaling Networks

Cell, organoid, or tissue models expressing RTKs activated by mutation, amplification, overexpression, and/or ligand engagement can be modulated by pharmacological inhibitors, RNAi knockdown, or CRISPR knockout. Time and the degree of knockdown are often explored. Validation of RTK impacts by immunoblot measuring RTK phosphorylation and downstream signaling often precede global mass spectrometry investigation of signaling networks.

Phosphorylation-dependent networks can be explored by phosphoprotein and phosphopeptide enrichment prior to analysis. Phosphotyrosine comprises only ~3% of cellular protein phosphorylation, where the use of the anti-phospho-tyrosine (anti-pY) antibody capture approaches of [8,10,11,12,13,14], which substantially enrich pY-containing proteins and peptides, has a marked advantage. The isolation of phospho-serine-, phospho-threonine-, and phospho-tyrosine-containing peptides has been performed most typically using the TiO_2_ bead or iron/gallium metal chelate bead affinity approaches [15,16,17,18,19]. In parallel, the isolation of cell, organoid, or tissue RNA followed by RNAseq or microarray can help define transcription networks impacted by modulation of the RTK function and downstream signaling. Bioinformatic approaches to infer signaling network changes from RNA abundance data have improved in the last decade, where the co-correlation of protein, phosphopeptide, and RNA data changes provides additional insights. A representative schema is illustrated in Figure 2.

### 3.2. MET Co-Option of Parallel Pathways in MET-Amplified Cells

One RTK may also positively affect the activity of another in a process that can be termed receptor co-option. Here, one dominant ligand-stimulated RTK, possibly amplified or mutated, directly or indirectly tyrosine phosphorylates additional RTKs to create functional signaling scaffolds and to engage signaling networks beyond the normal capabilities of the original dominant kinase. Where the co-option of RTK signaling networks is observed, markers of RTK activation (e.g., the extent of receptor tyrosine phosphorylation) are not necessarily predictive of onco-addiction. Direct or indirect crosstalk among EGFR, ERBB2/HER2, MET, IGF1R, PDGFR, and other RTKs has been observed in a bidirectional manner, recently reviewed by Paul and Hristova [20]. Cross phosphorylation of RTKs in trans can allow signaling scaffolds to be established, mimicking kinase activation in the absence of a ligand.

The MET oncoprotein is a heteromeric, single-membrane-spanning, type I receptor tyrosine kinase. It comprises two subunits, 45 kDa and a 145 kDa disulfide-linked, and is activated by hepatocyte growth factor (HGF). MET is amplified in some solid tumors, for example, NSCLC with an incidence of ~4%. MET also can be mutationally activated, notably by exon 14 skipping in NSCLC, and second generation MET inhibitors were recently FDA-approved for that indication.

The co-option of diverse kinase signaling networks by a single RTK is observed in the NSCLC H1993 cell line with amplified MET, the receptor for hepatocyte growth factor (HGF). Exposure to small-molecule MET tyrosine kinase inhibitor crizotinib attenuated both MET and MET-associated SH2/PTB domain adapter proteins as well as more distantly related RTK signaling networks two hours after drug treatment (Table 1). As expected, the MET inhibitor crizotinib (1 μM, 2 h) rapidly decreased the tyrosine phosphorylation of MET and its major activating autophosphorylation motif at positions Y1234 and Y1235. Interestingly, the inhibition of MET also inhibited the tyrosine phosphate content on the ‘autophosphorylation’ sites of EGFR (e.g., Y1173) and major activating tyrosine phosphorylation sites on ITGB4, ITGA6, EPHA2, and EPHB4 (Table 1 and data not shown). Consistent with the co-option of the EGFR by MET, the exposure of H1993 cells to the EGFR inhibitor erlotinib (1 μM, 2 h) failed to reduce tyrosine phosphorylation of EGFR or ErbB2/HER2. The co-option of the EGFR and ErbB2 by MET was verified by Western immunoblot, as shown in Figure 3A.

This attenuation of MET activation by crizotinib then results in a comparatively complete dephosphorylation of a wide array of signaling adaptors, cell–cell junction proteins, cytoskeletal reorganizing elements, and folding chaperones (data not shown). In this amplified MET cell model, MET is a major source of phosphotyrosine signaling, where MET essentially ‘highjacks’ other RTKs as signaling adapters to increase the size and scope of its signaling network, as illustrated in Figure 3B. The interaction of MET and the EGFR has been actively studied as a source of resistance to EGFR inhibitors in mutant EGFR NSCLC [21,22,23,24]. The physical interactions of MET and the EGFR have been elucidated using advanced FRET and microscopy approaches [25,26], where MET-EGFR dimerization was promoted by EGFR mutation [26].

### 3.3. Reciprocal Activation or Inhibition of IGF1R and IR

The insulin-like growth factor receptor (IGF1R) is a 155kDa membrane receptor tyrosine kinase which binds insulin-like growth factor-1 (IGF1) and, at lower affinity, IGF2 and insulin. It is synthesized as a preformed tetramer comprising two alpha and beta chains linked by disulfide linkages, reviewed by [27,28]. IGF1 binding results in IGF1R kinase activation, tyrosine autophosphorylation, and phosphotyrosine (pY)-dependent binding of critical signaling adaptors IRS1, IRS2, SHC1, and other SH2 or PTB domain pY-binding proteins, ultimately resulting in PI3K-AKT and RAS pathway activation. IGF1R signaling plays a crucial role in controlling biological outcomes, such as cellular growth, proliferation, differentiation, survival against apoptosis, and migration [29]. Notably, in breast cancer, IGF1R activation promotes cell survival and is implicated in oncogenic progression [30]. In the Ewings sarcoma line A673, IGF-1R kinase inhibition resulted in the reciprocal activation of the insulin receptor and cell survival [31]. Similarly, reciprocal activation of the EGFR by IGF1R inhibition and activation of IGF1R through EGFR inhibition has been observed in the colorectal GEO cell and in A673 cell models. Mass spectrometry has been used in defining signaling changes in GEO cells exposed to EGFR inhibitors (1 μM, 2 h), where the insulin receptor (IR) was reciprocally activated by the EGFR. Anti-pY antibody affinity steps were used to isolate phosphotyrosine-containing proteins and iTRAQ stable isotope-labelling was used for relative quantitation in untargeted mass spectrometry experiments, as shown in Table 2.

Crosstalk among EGFR, MET, IGF1R, and IR was measured in the colorectal model line GEO, which expresses all four RTKs. Phosphorylation signaling networks sensitive to IGF1R or EGFR inhibition were initially investigated by phospho-profiling mass spectrometry using stable isotope coupled with a phosphotyrosine antibody capture. Inhibition of EGFR tyrosine kinase activity with erlotinib reduced tyrosine phosphorylation of MET, indicating EGFR was co-opting MET as a signaling scaffold. This was observed by phospho-profiling mass spectrometry (Table 2), where EGFR inhibition with erlotinib (1 μM, 2 h) decreased tyrosine phosphate content both for MET (−5.1 log_2_-fold change (FC); *p* < 0.0001) and for EGFR (log_2_-FC −2.5; *p* = 0.007). Interestingly, inhibition of EGFR reproducibly increased tyrosine phosphorylation of the insulin receptor (IR) (log_2_-FC 1.5, though statistical significance was not obtained). EGFR inhibition resulted in a comparable increase in phospho-IGF1R, and EGFR inhibitor increases in both the IR and IGF1R were observed by immunoblot using IR and IGF1R phospho-specific antibodies [31,32]. Exposure of GEO cells to a dual IGF1R/IR kinase inhibitor (linsitinib, 1 μM, 2 h) resulted in an expected attenuation in IR phosphotyrosine content (log_2_-FC −1.9). However, cell adaptation to RTK inhibition can be observed. Forty-eight hours post-EGFR addition, increased EGFR, MET, and IR phosphorylation can be observed, which may be dependent, in part, on FOXO nuclear translocation and transcriptional activation downstream of AKT inhibition ({Chandarlapaty, 2011 #53}. This shift from reduced to increased pY content between the 2 and 48 h timepoints was also observed in the SH2 domain RTK-signaling adaptor GRB2 and in the p85 PI3K subunit, both proteins which can directly or indirectly interact with activated RTKs. In contrast, the inhibition of the Ephrin A2 receptor and MAPK phosphotyrosine content was similar between the 2 and 48 h time points. The dual IGF1R/IR inhibitor (linsitinib, 1 μM, 48 h) also increased the pY content in EGFR, MET, and GRB2 after 48 h (Table 2) but further inhibited IR (log_2_-FC −4.1, *p* < 0.001). The complete inhibition of pY-IRS-1 as well as downstream signaling through the MAPK and AKT pathways was achieved only when IGF-1R and IR were co-inhibited, highlighting the need for dual IGF1R/IR inhibitors in cancer indication [31].

The mechanisms by which RTK reciprocity is achieved are still not well described, although relief of the negative feedback of S6 kinase (S6K) on IRS1 is thought to play a critical role in IGF1R and IR crosstalk [32]. The ability of tumor cells to reciprocally induce alternative RTKs, as illustrated in Figure 4, following the inhibition of a given receptor (in this case, the EGFR or IGF1R), highlights the need for a rational combination of anti-cancer therapy.

The mechanisms by which RTK reciprocity is achieved are still not well described, although relief of the negative feedback of S6 kinase (S6K) on IRS1 is thought to play a critical role in IGF1R and IR crosstalk [27]. The ability of tumor cells to reciprocally induce alternative RTKs, following the inhibition of a given receptor (in this case, the EGFR or IGF1R), highlights the need for a rational combination of anti-cancer therapy.

### 3.4. EGFR Co-Option of Tyrosine Kinase Pathways

The epidermal growth factor receptor (EGFR) RTK is an important transducer of cell survival, proliferation, and migratory signals during fetal development and in adult normal and tumor cell homeostasis [1]. The EGFR is a receptor tyrosine kinase of ~170 kDa, which is activated by the EGF family of growth factors (e.g., EGF, TGFA, AREG, EREG, BTC, and HBEGF) and EGF-domain-containing proteins [33]. Inhibitors of the EGFR are used in the treatment of colorectal, head, neck, and non-small-cell lung cancers [34], and EGFR inhibitors have shown activity in other tumor types, for example, in subsets of breast and bladder cancer with EGFR activation. In carcinomas with an epithelial phenotype, onco-addiction and tumor progression have been associated with EGFR amplification, with autocrine-ligand-dependent activation of EGFR through genetic mutations. For example, in NSCLC adenocarcinomas, mutations in exons 19–21 or by point mutation of L858R in EGFR [35,36,37], which alter ATP binding [38], are observed in 15% (European) to 35% (Asia) of cases. In glioblastoma (GBM), EGFR amplification and losses of exons 2–7 are associated with partial EGFR dependence [39].

In non-small-cell lung cancer (NSCLC), patients with activating EGFR mutations are treated with EGFR inhibitors as a first-line therapy. However, eventual resistance to anti-EGFR therapies are common, and multiple resistance mechanisms have been described. Resistance to EGFR inhibitors can occur, mainly in these patients with these primary activating mutations, from acquisition of a secondary mutation of EGFR at the T790M gatekeeper site [40,41], which increases ATP affinity [38]. Secondly, resistance can also occur through upregulation of alternate receptor tyrosine kinases (RTKs) [42]. Thirdly, resistance can also occur through epigenetic cell plasticity, notably through epithelial mesenchymal transition (EMT) [43,44] or through a related small-cell-like neuroendocrine trans-differentiation, where, in both cases, alternative survival and migratory signaling pathways are engaged.

Resistance can derive from over-activation of partially redundant pathways concurrently active in a cell. For example, activation of the IGF-1 receptor (IGF-1R) or the HGF receptor (MET) has been shown to obviate the need for EGFR signaling in epithelial-derived lung tumors [22]. Crosstalk has been well documented with EGF, HGF, and IGF1 receptors. These receptors, when activated by ligand-binding, can create network redundancies (for example, by IGF stimulation of EGFR onco-addicted cells [8]). Similarly, HGF has been correlated with EGFR resistance in patient specimens [45].

In the NSCLC line H292, ~50% of the tyrosine phosphate on MET is inhibited by the EGFR blockade. Crosstalk between the EGFR and MET was measured by combined anti-pY affinity selection, stable isotope peptide-labeling (iTRAQ), and LC-electrospray tandem MS. The tyrosine phosphorylation of MET was decreased following 1 and 4 h exposure to erlotinib (2 μM) relative to mock control cells (Table 3).

The inhibition of phospho-MET followed kinetics for the erlotinib-dependent decrease in the SH2 adapter proteins phospho-SHC, -MAPK1, and -GRB2. The data suggest direct phosphorylation of MET by the EGFR or a rapid recruitment and activation of intermediary non-transmembrane tyrosine kinases, as illustrated in Figure 5. In support of direct MET-EGFR interaction, direct visualization of MET-EGFR interactions have been measured by spectroscopy approaches [26].

### 3.5. Reciprocal PDGFR—EGFR Activation in EMT-Derived Cells

Tumor tissues progress from in situ to metastatic states through the reacquisition of developmental programs, allowing invasion and metastasis. The acquisition of an invasive phenotype can occur by epithelial–mesenchymal transition (EMT). The molecular characteristics of epithelial and mesenchymal cell phenotypes were extensively characterized by intersection of proteomic, phosphoproteomic, and gene expression profiling approaches. Tumor cells that have undergone EMT show a marked reduction in sensitivity to EGFR TKIs [43] and anti-EGFR MAbs. In several instances, the EMT-derived mesenchymal-like tumor cells have gained sensitivity to PDGFR and/or FGFR1 inhibitors. Significant switching of receptor tyrosine kinases, from EGFR, MET/Ron, and IGF-1R to cells utilizing PDGFR and FGFR was observed. The acquisition of autocrine fibronectin–integrin was also observed in several tumor lines and inducible models. Therefore, EMT can promote use of alternative signaling pathways and shift autocrine RTK activation to more pronounced stromal cell paracrine RTK activation.

Reciprocal activation of PDGFRα was observed when EGFR was inhibited in the NSCLC line H1703, as illustrated in Figure 6 [42]. These studies involved a quantitative anti-phosphotyrosine profiling coupled with the global LC-MS/MS shotgun described in Materials and Methods and shown in Table 4.

The PDGFR-amplified H1703 cell model was treated with EGF (10 ng/mL) and either EGFR inhibitor erlotinib (5 μM) or DMSO control (0.2%), followed by anti-pY phosphoprotein isolation, digestion, and iTRAQ peptide-labeling. EGFR inhibition resulted in a modest but statistically significant increase in tyrosine phosphorylation of PDGFRA, and associated signaling adapters (PI3KCA, PLCγ1) were observed at 1 and 4 h versus the DMSO vehicle control. In contrast, EGFR inhibition decreased the tyrosine phosphorylation of CBL and SHC1, suggesting specific recruitment by the EGFR. The erlotinib-mediated increase in phospho-PDGFR was confirmed by Western analysis using anti-PDGFR Y754- and Y720-specific antibodies, quantitated by densitometry (Figure 6).

## 4. Conclusions

Through RTK co-option, RTKs can expand their signaling scaffolds of non-family RTKs to enhance aberrantly the network activation. The measurement of RTK ‘auto’ phosphorylation may not always be an indicator of intrinsic kinase activity. Reciprocal RTK activation can be a source of resistance to RTK and downstream inhibitors, where EGFR inhibition can activate the IR, and EGFR inhibition can activate PDGFRα. The quantitative measurement of cancer cell signaling under dynamic conditions of pharmacological or siRNA-mediated inhibition of specific signaling nodes provides insight into the requirements for the effective cellular blockade of survival and invasive networks associated with cancer progression. The use of quantitative shotgun LC-MS/MS methods incorporating stable isotopes provides a rapid means to identify proteins and phosphoproteins, verifiable by scheduled MRM or immunoblot. These methods have allowed interrogation of RTK crosstalk in tumor cell lines and xenografts to generate specific hypotheses related to targeted drug combinations.

## Figures and Tables

**Figure 1 proteomes-11-00020-f001:**
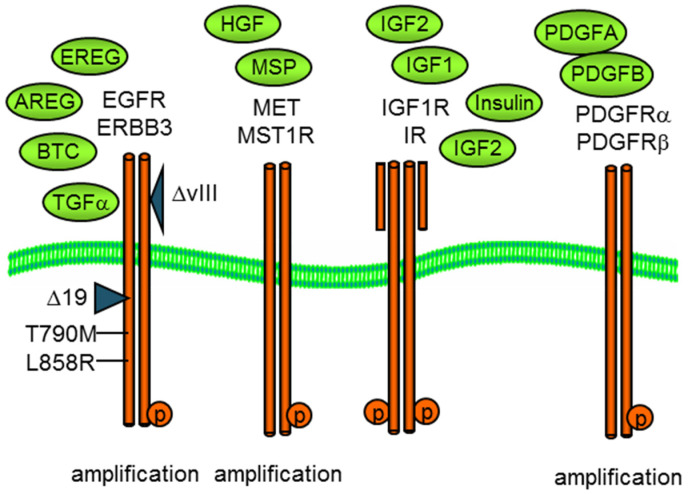
Growth factor activation of membrane receptor tyrosine kinases, EGFR, MET, IGF1R, and PDGFR as well as related kinases. Both oncogenic EGFR and MET can exhibit copy number amplification and/or mutational activation, exon 19, or L858R for EGFR or exon 14 for MET.

**Figure 2 proteomes-11-00020-f002:**
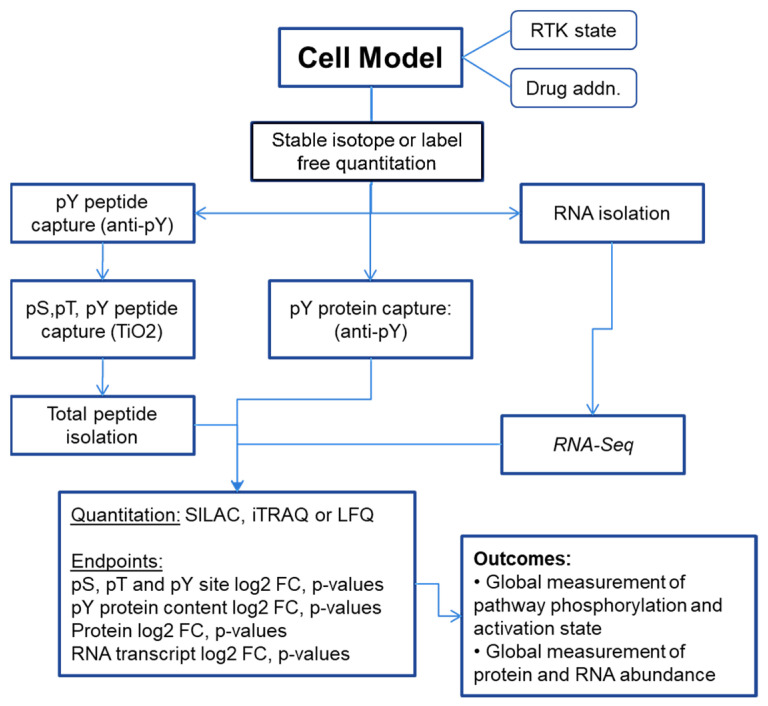
Representative schema for measuring RTK signaling networks from cell, organoid, or tissue models.

**Figure 3 proteomes-11-00020-f003:**
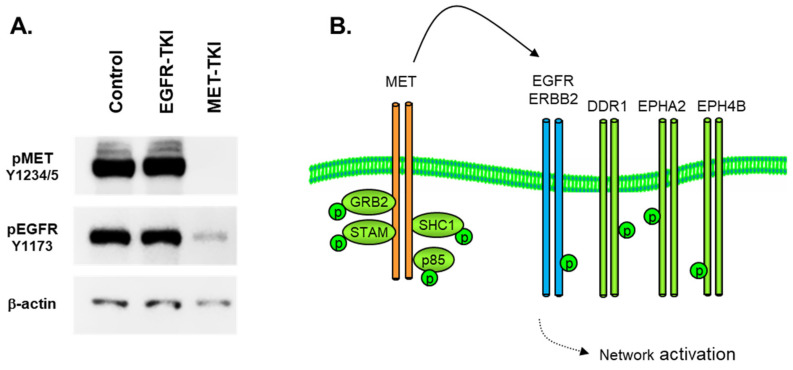
(**A**) Immunoblot in H1993 MET-amplified NSCLC cells, showing inhibition of MET (Y1234, Y1235) and EGFR (Y1173) tyrosine phosphorylation by MET-TKi (crizotinib, 6 h, 1 μM) but no reduction in EGFR (Y1173) with EGFR-TKi (erlotinib, 6 h, 1 μM). Phospho-specific antibodies were used as described in Materials and Methods.β-actin was used as a loading control. (**B**) MET-dependent activation of parallel signaling pathways by RTK co-option.

**Figure 4 proteomes-11-00020-f004:**
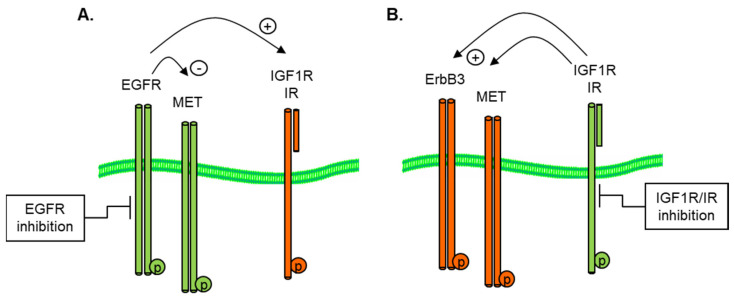
Reciprocal activation of (**A**) IGF1R by EGFR inhibition and (**B**) EGFR by IGF1R/IR inhibition in GEO CRC and A673 Ewings cell models.

**Figure 5 proteomes-11-00020-f005:**
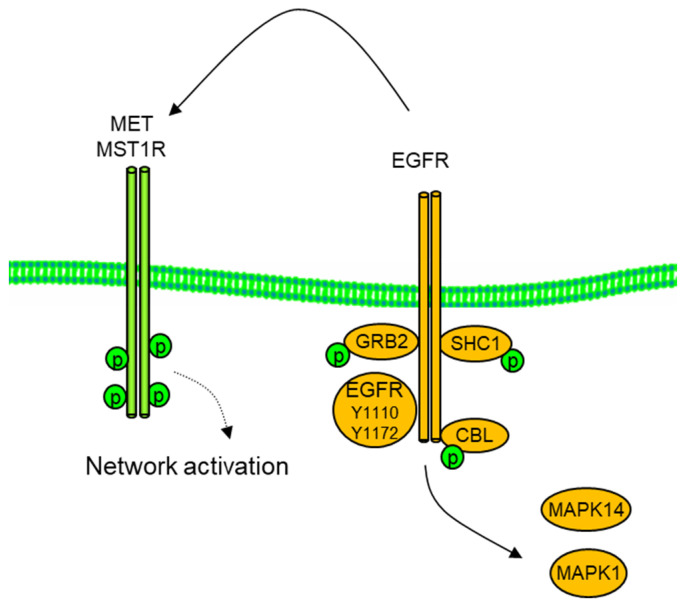
Co-option of MET by EGFR in H292 NSCLC cells. Inhibition of EGFR and downstream SH@ domain adaptors GRB2, SHC1, CBL, and MAPK1/Erk2 with erlotinib (1 μM, 2 h) resulted in reduced MET and RON/MST1R phospho tyrosine.

**Figure 6 proteomes-11-00020-f006:**
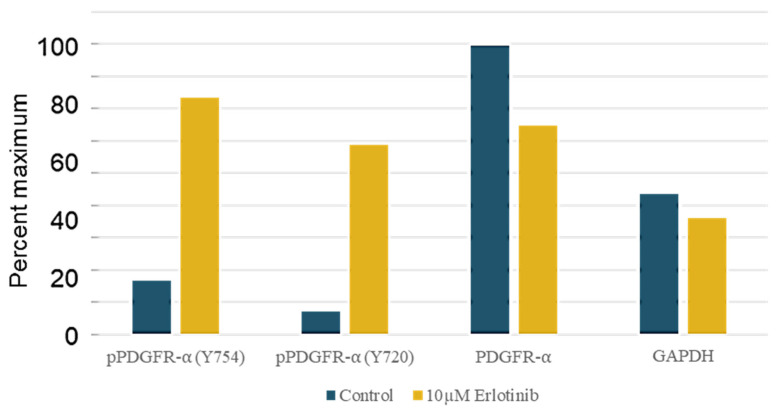
Western immunoblot quantitation shows EGFR inhibitor erlotinib (10 μM, 2 h) increases PDGFRα tyrosine phosphorylation in H1703 NSCLC cells. Antibodies to phospho-PDGFRα (Y754, Y720), total PDGFRα, and GAPDH-loading control were used as described in Materials and Methods. Densitometry values are expressed as percent of the maximum signal (here, PDGFRα Control).

**Table 1 proteomes-11-00020-t001:** Protein phosphotyrosine content in MET-amplified NSCLC H1993 cells measured by pY affinity capture, iTRAQ stable isotope-labeling, and untargeted LC-MS/MS. Log_2_-fold change ratios comparing MET (1 μM METi crizotinib, 2 h) or EGFR (1 μM EGFRi erlotinib, 2 h) inhibitors with DMSO control and associated *p*-values are shown.

Acc	Gene	Protein	METi 2 h	p METi	EGFRi 2 h	p EGFRi
P08581	MET	Hepatocyte growth factor receptor	−4.22	0.000	0.38	0.000
	Y1234,Y1235	DMYDKEpYpYSVHNK (n = 86 spectra, 99% confidence)	−3.13	0.000	0.10	ns
P00533	EGFR	Epidermal growth factor receptor	−3.11	0.000	0.48	0.000
	Y1197	GSTAENAEpYLR (n = 45 spectra, 99% confidence)	−4.16	0.000	0.05	ns
P04626	ERBB2	Receptor tyrosine–protein kinase erbB-2	−2.23	0.013	0.41	0.020
Q08345	DDR1	Epithelial discoidin domain-containing receptor 1	−3.55	0.000	0.49	0.001
P29317	EPHA2	Ephrin type-A receptor 2	−2.54	0.000	0.28	0.031
P54760	EPHB4	Ephrin type-B receptor 4	−1.62	0.048	0.42	ns
Q96RT1	ERBIN	Protein LAP2	−3.27	0.000	0.16	0.019
P62993	GRB2	Growth factor receptor-bound protein 2	−2.95	0.000	0.26	0.004
	Y209	NpYVTPVNR (n = 29 spectra, 99% confidence)	<−3.32	0.000	0.13	ns
P29353	SHC1	SHC-transforming protein 1	−2.42	0.012	0.37	0.044
	Y349,Y350	MAGFDGSAWDEEEEEPPDHQpYpYNDFPGKEPPLGGVVDMR	−3.29	0.010	0.39	ns
Q06124	PTN11	Tyrosine–protein phosphatase type 11	−3.44	0.022	0.33	ns

**Table 2 proteomes-11-00020-t002:** Activation of IGF1R by EGFR inhibition in GEO CRC cells after two hours of erlotinib (1 μM) exposure. IGF1R/IR inhibitor (linsitinib 1 μM) maintained a reduction in IGF1R and IR tyrosine phosphorylation after 48 h. In contrast, EGFR and MET showed marked increases in tyrosine phosphorylation after 48 h, suggesting feedback-mediated rebound. Log_2_-fold change and associated *p*-values are shown.

Acc	Gene	Protein	Time (h)	EGFRi	IGF1Ri/IRi	p EGFRi	p IGF1Ri/IRi
P00533	EGFR	Epidermal growth factor receptor	2	−2.47	−0.40	0.007	ns
			48	1.24	1.41	0.000	0.000
P29317	EPHA2	Ephrin type-A receptor 2	2	−1.38	−0.05	ns	ns
			48	−2.52	−0.57	0.000	0.004
P62993	GRB2	Growth factor receptor-bound protein 2	2	−4.01	−0.86	0.010	ns
			48	1.08	1.18	0.002	0.000
P06213	INSR	Insulin receptor	2	1.49	−1.91	ns	ns
			48	2.02	−4.08	0.000	0.000
P28482	MAPK1	Mitogen-activated protein kinase 1	2	−1.18	0.21	ns	ns
			48	−2.63	−1.51	0.001	0.006
P08581	MET	Hepatocyte growth factor receptor	2	−5.01	−0.27	0.000	0.131
			48	1.18	1.87	0.000	0.000
P27986	PIK3R1	Phosphatidylinositol 3-kinase regulatory a	2	−1.99	−0.85	0.002	0.038
			48	1.63	0.53	0.000	0.002

**Table 3 proteomes-11-00020-t003:** Co-option of MET by EGFR inhibition in NSCLC H292 cells. EGFR inhibition (erlotinib 1 μM for 1, 4, or 24 h) was measured by anti-pY affinity chromatography and iTRAQ stable isotope-labeling. Specific EGFR inhibition with erlotinib correlated with MET and MST1R/Ron inhibition. Log_2_-fold change and associated *p*-values are shown.

Acc	Gene	Protein	EGFRi 1 h	p 1 h	EGFRi 4 h	p 4 h	EGFRi 24 h	p 24 h
P00533	EGFR	Epidermal growth factor receptor	−0.61	0.000	−0.62	0.000	−0.90	0.000
P08581	MET	Hepatocyte growth factor receptor	−0.96	0.001	−0.63	0.126	−0.89	0.000
Q04912	MST1R	Macrophage-stimulating protein receptor	−1.28	0.000	−1.51	0.001	−1.24	0.000
Q06124	PTN11	Tyrosine–protein phosphatase 11	−1.77	0.025	−2.34	0.057	−1.57	0.009
P62993	GRB2	Growth factor receptor-bound protein 2	−1.29	0.000	−2.66	0.002	−1.77	0.000
P22681	CBL	E3 ubiquitin–protein ligase CBL	−2.17	0.000	−3.38	0.000	−2.84	0.000
P29353	SHC1	SHC-transforming protein 1	−1.99	0.000	−2.22	0.000	−1.69	0.000
O75886	STAM2	Signal-transducing adapter molecule 2	−1.49	0.038	−5.83	0.032	−2.83	0.021
O14964	HGS	HGF-regulated substrate	−1.42	0.000	−4.46	0.000	−1.55	0.000
P28482	MAPK1	Mitogen-activated protein kinase 1	−1.70	0.000	−2.42	0.000	−1.41	0.000
Q16539	MAPK14	Mitogen-activated protein kinase 14	−0.82	0.001	−0.79	0.013	−0.73	0.000

**Table 4 proteomes-11-00020-t004:** H1703 cells treated with EGF (10ng/mL) and EGFRi erlotinib (5 μM) show modest but significant upregulation of tyrosine phosphorylation of PDGFR. Log_2_-fold change and associated *p*-values are shown.

Acc	Gene	Protein	EGF + EGFRi 1 h	p 1 h	EGF + EGFRi 4 h	p 4 h
P16234	PDGFRA	Platelet-derived growth factor receptor a	0.27	0.000	0.08	0.016
P27986	PIK3R2	PI3-kinase p85-subunit a	0.25	0.007	0.32	0.001
O00459	PIK3R1	PI3-kinase p85-subunit b	0.49	0.036	0.24	0.189
P42336	PIK3CA	PI3-kinase p110 subunit a	0.65	0.003	0.53	0.008
P42338	PIK3CB	PI3-kinase p110 subunit b	0.41	0.139	0.44	0.108
P19174	PLCG1	Phospholipase C-g	0.34	0.000	0.26	0.000
Q06124	PTPN11	Tyrosine–protein phosphatase 11	0.38	0.000	0.22	0.000
Q07889	SOS1	Son of sevenless homolog 1	0.34	0.102	0.60	0.037
O43639	NCK2	Cytoplasmic protein NCK2	0.55	0.013	0.61	0.002
Q13153	PAK1	Serine/threonine–protein kinase PAK 1	0.64		0.69	
Q15052	ARHGEF6	Rho guanine nucleotide exchange factor 6	0.74	0.000	1.05	0.000
P22681	CBL	E3 ubiquitin–protein ligase CBL	−1.34	0.000	−1.23	0.000
P29353	SHC1	SHC-transforming protein 1	−1.26	0.000	−1.38	0.000

## Data Availability

Complete data tables are available upon request. Complete phosphoprotein and phosphopeptide data from H1993, GEO, H292, and H1703 cells will be submitted to MassIVE.

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
