# Peer review of "Mass Spectrometry and Pharmacological Approaches to Measuring Cooption and Reciprocal Activation of Receptor Tyrosine Kinases"

_proteomes, 2023, doi:10.3390/proteomes11020020_

Round 1

Reviewer 1 Report

In the present manuscript, authors have explored " Mass spectrometry and pharmacological approaches to measuing receptor tyrosine kinase signaling crosstalk". The study lacks a specific hypothesis being tested.

·         Title doesn’t reflect the essence of the work       

·         Abstract not in sound, rewrite with more results

·         Improve the quality of all the figures

·         Introduction is not sufficient, it should be rewritten with proper referencing, no reference was cited in introduction.

Moderate editing of English language is required.

Author Response

In the present manuscript, authors have explored " Mass spectrometry and pharmacological approaches to measuring receptor tyrosine kinase signaling crosstalk". The study lacks a specific hypothesis being tested. We test the hypothesis that co-option and reciprocal regulation of receptor tyrosine kinases occur in cancer cells and can be measured by combination of mass spectrometry and pharmacological inhibitors.

Title doesn’t reflect the essence of the work. We changed the title to focus on receptor tyrosine kinase cooption and reciprocal activation (comprehensive signaling crosstalk would be beyond the scope of this study)

Abstract not in sound, rewrite with more results. We agree, the abstract was re-written to reflect the key findings.

 Improve the quality of all the figures. Figure 3 was modified and Western blot confirmation provided. Figure 5 was modified. Figure 7 showing Western blot quantitation for PDGFR induction was corrected to include a y-axis label.

Introduction is not sufficient, it should be rewritten with proper referencing, no reference was cited in introduction. Agreed, references and additional text were added to the Introduction section.  Background and specific points related to MET, EGFR, IGF1R/IR and PDGFR are individually made in the Results/Discussion section. The Introduction establishes a general background and framework for the subsequent sections where RTK-specific details are described. Overall the manuscript is part Article and part Review, suited to the Proteomes special issue theme in ‘Proteomics in Cancer Research’.

Reviewer 2 Report

The article "Mass spectrometry and pharmacological approaches to measuring receptor tyrosine kinase signaling crosstalk" explains the measurement of crosstalk among Receptor Tyrosine Kinase (RTK)-dependent signaling pathways using mass spectrometry. However, the article's clarity is limited because it focuses more on measurements than the factors responsible for their effectiveness.

- There is recent evidence that alterations in signaling pathways can occur through specific or nonspecific interactions, which can lead to false representations of disease conditions. Additionally, GPCRs are important factors in altering these pathways and directly responsible for crosstalks that are not mentioned in the article.

Reference:

Sanmukh, S.G., Dos Santos, N.J., Barquilha, C.N., De Carvalho, M., Dos Reis, P.P., Delella, F.K. ... Felisbino, S.L. (2023). Bacterial RNA virus MS2 exposure increases the expression of cancer progression genes in the LNCaP prostate cancer cell line. Oncology Letters, 25, 86. https://doi.org/10.3892/ol.2023.13672

- Even if the authors are primarily interested in targeting RTKs, it's important to consider all possible targets, including those reported by the authors, when examining crosstalks.

- Improving the article's discussion by exploring the factors responsible for crosstalks in the introduction and discussion, as well as performing gene co-expression analysis, would greatly enhance the overall quality of the article.

- In addition, interaction studies of the reported proteins could provide valuable information on crosstalks and RTK signaling.

The English language in this article is fine except few grammatical errors!

Author Response

The article "Mass spectrometry and pharmacological approaches to measuring receptor tyrosine kinase signaling crosstalk" explains the measurement of crosstalk among Receptor Tyrosine Kinase (RTK)-dependent signaling pathways using mass spectrometry. However, the article's clarity is limited because it focuses more on measurements than the factors responsible for their effectiveness. We added text and references regarding the mechanisms of RTK activation: RTK mutations, gene amplication, RTK overexpression and ligand independent activation, and autocrine and paracrine growth factor activation.

There is recent evidence that alterations in signaling pathways can occur through specific or nonspecific interactions, which can lead to false representations of disease conditions. Additionally, GPCRs are important factors in altering these pathways and directly responsible for crosstalks that are not mentioned in the article. We have changed the title to reflect a focus on RTK cooption and reciprocal activation. A detailed discussion of the crosstalk of RTK downstream signals becomes more and more complicated with time following perturbation, and is beyond the scope of this manuscript.

Even if the authors are primarily interested in targeting RTKs, it's important to consider all possible targets, including those reported by the authors, when examining crosstalks. Comprehensive examination of protein interactions and all signaling targets is beyond the scope of the four RTKs (MET, EGFR, IR/IGF1R and PDGFR) investigated. For example, the physical and functional interactions between GPCRs and RTKs, or mucin family members and RTKs are broad subject areas beyond the scope of this manuscript. Covering all targets intersecting RTK signaling again is beyond a single manuscript.

Improving the article's discussion by exploring the factors responsible for crosstalks in the introduction and discussion, as well as performing gene co-expression analysis, would greatly enhance the overall quality of the article. In addition, interaction studies of the reported proteins could provide valuable information on crosstalks and RTK signaling. We have added text and references in the Introduction and Discussion to enhance the factors contributing to crosstalk, but have focused the manuscript to a greater extent on cooption and reciprocal activation between RTKs.

Reviewer 3 Report

The manuscript's authors describe how they used quantitative shotgun mass spectrometry to investigate receptor tyrosine kinase (RTK) cross-talk in cancer cell lines where targeted RTK signaling pathways co-opt other signaling pathways that may interfere with anti-cancer therapies. Anti-phosphotyrosine affinity fractions were isolated and analyzed by LC-MS/MS. They find inhibition of EGFR leads to IR activation as well as PDGFR. 

Overall the manuscript is interesting, but some issues need to be addressed before it is suitable for publication.

1. The results lack data confirming the mass spec shotgun results. Confirmation of mass spec results by Western blotting for each experiment should be shown by immunoprecipitation with pTyr antibodies and blotting with antibodies to the proteins listed in the tables that were captured by isotope labeling. 

2. The figure legends are only titles and should include descriptions of the figures.

3. The graph in Figure 7 lacks X and Y-axis labels and no description in the figure legend. 

Author Response

The results lack data confirming the mass spec shotgun results. Confirmation of mass spec results by Western blotting for each experiment should be shown by immunoprecipitation with pTyr antibodies and blotting with antibodies to the proteins listed in the tables that were captured by isotope labeling. Western data provided for RTK cooption (Figure 3) and RTK reciprocal activation (Figure 7). We reference our published Western confirmation for insulin receptor activation by EGFR inhibition. In global proteomic studies we use statistics to establish significant protein or phosphopeptide changes. In our previous studies (eg. Thelemann, A., Petti, F., Griffin, G., Iwata, K., Hunt, T., Settinari, T., Fenyo, D., Gibson, N. and Haley, J.D. (2005) ‘Phosphotyrosine signaling networks in epidermal growth factor receptor overexpressing squamous carcinoma cells’ Mol. Cell. Proteomics 4, 356-376) we did use Western analysis to validate changes identified using global mass spectrometry and stable isotope labeling. However advances in mass spectrometry instrumentation and computer analyses make Western confirmation less common, where is it sometimes used for key findings.

The figure legends are only titles and should include descriptions of the figures. We added needed detail to the Figure legends.

The graph in Figure 7 lacks X and Y-axis labels and no description in the figure legend.  Figure 7 and its legend were corrected.

Reviewer 4 Report

Nice work. I don't have any major comment about it. I just think authors need to upload the MS raw data and the Uniprot dataset to public database. 

Author Response

Nice work. I don't have any major comment about it. I just think authors need to upload the MS raw data and the Uniprot dataset to public database. Yes, we will upload H1993, H292, GEO, H1703 data to MassIVE.

Round 2

Reviewer 3 Report

The methods is missing the description of Western blotting results from Figure 3A.

Figure 3A has no loading control for each receptor. Was the blot probed with a pan-phospho-Tyrosine antibody or antibodies specific to a phospho-residue on each specific receptor.

Figure 7 still does not provide enough information to tell the reader what they are looking at. "Percent maximum" of WHAT? What antibodies were used in this experiment?? Three different specific phospho-antibodies? Unclear. 

Line 246 page 7 the reference formatting is not consistent.

Line 126 page 3 Peptide identification-tions??

Author Response

We completely agree and thank the reviewer for noticing these important omissions. 

In the Methods section we included new text on antibodies and immunodetection: Antibodies directed to phospho-MET (Y1234, Y1235; #3077S), phospho-EGFR (Y1173S; #4407), beta-actin (#8457S), and PDGFRa (#3164S) were from Cell Signaling Technologies. Antibodies to pPDGFR (Tyr 754, #12911), pPDGFR (Tyr720, #12910) and GAPDH (#25778) were from Santa Cruz Biotechnology. Protein immunodetection was performed by electrophoretic transfer of SDS-PAGE separated proteins to PVDF, incubation with antibody and chemiluminescent second step detection.

In Figure 3A we included a new Western blot with beta-actin control. We included a new Figure 3A legend describing the phospho-specific antibodies and conditions used: Immunoblot in H1993 MET amplified NSCLC cells, showing inhibition of MET (Y1234, Y1235) and EGFR (Y1173) tyrosine phosphorylation by MET-TKi (crizotinib, 6hrs, 1uM), but no reduction in EGFR (Y1173) with EGFR-TKi (erlotinib, 6hrs, 1uM). Phospho-specific antibodies were used as described in Materials and Methods. b-actin was used as a loading control. 

We revised Figure 7 X-axis labels and better defined the y-axis in the new legend: Western immunoblot quantitation shows EGFR inhibitor erlotinib (10 uM, 2 hrs) increases PDGFRa tyrosine phosphorylation in H1703 NSCLC cells. Antibodies to phospho-PDGFRa (Y754, Y720), total PDGFRa and GAPDH loading control were used as described in Materials and Methods. Densitometry values are expressed as percent of the maximum signal (here PDGFRa Control).